# A KERNEL RANDOM MATRIX-BASED APPROACH FOR SPARSE PCA

**Mohamed El Amine Seddik**[1,2], **Mohamed Tamaazousti**[1] **& Romain Couillet**[2,3,*]
[1]CEA List, [2]CentraleSupélec, [3]GIPSA-Lab University of GrenobleAlpes
{mohamedelamine.seddik,mohamed.tamaazousti}@cea.fr
romain.couillet@gipsa-lab.grenoble-inp.fr

## ABSTRACT

In this paper, we present a random matrix approach to recover sparse principal components from $n$ $p$-dimensional vectors. Specifically, considering the large dimensional setting where $n, p \to \infty$ with $p/n \to c \in (0, \infty)$ and under Gaussian vector observations, we study kernel random matrices of the type $f(\hat{\mathbf{C}})$, where $f$ is a three-times continuously differentiable function applied entry-wise to the sample covariance matrix $\hat{\mathbf{C}}$ of the data. Then, assuming that the principal components are sparse, we show that taking $f$ in such a way that $f'(0) = f''(0) = 0$ allows for powerful recovery of the principal components, thereby generalizing previous ideas involving more specific $f$ functions such as the *soft-thresholding* function.

## 1 INTRODUCTION

Principal component analysis (PCA) is extensively used in data analysis and machine learning applications. It is a dimension reduction technique that aims to project a given dataset onto principal subspaces spanned by the leading eigenvectors of the sample covariance matrix (Wold et al., 1987), which represent the principal modes of variance. Basically, the statistical interpretation of PCA lies in the fact that most of the variance in the data is captured by these modes. Consequently, PCA reduces the dimension of the feature space while keeping most of the information in the data. It is well-known (Anderson, 1963) that PCA performs efficiently in the traditional data setting where the number of features is small and the number of samples is large.

Consider a data matrix $Y \in \mathbb{R}^{p \times n}$ consisting of $n$ centered samples, each sample having $p$ features. The standard PCA method requires the computation of the sample covariance matrix $\hat{\mathbf{C}} = YY^\intercal/n$ and estimates the first principal components $u_1, u_2, \ldots$ (*i.e.*, the successive dominant eigenvectors of $\mathbf{C} = \mathbb{E}[YY^\intercal/n]$) by the ordered eigenvectors $\hat{u}_1, \hat{u}_2, \ldots$ of $\hat{\mathbf{C}}$. (Johnstone & Lu, 2009) demonstrated that, in the high dimensional regime where $n, p \to \infty$ with $p/n \to c > 0$, the principal component $\hat{u}_1$ estimated by standard PCA is inconsistent. Essentially, if $p/n \nrightarrow 0$ then $\|\hat{u}_1 - u_1\|_2 \nrightarrow 0$ in the high-dimensional asymptotic regime. This phenomenon is well investigated within the field of random matrix theory for covariance models of the form $\hat{\mathbf{C}} = \frac{1}{n}\Sigma_p^{1/2} X X^\intercal \Sigma_p^{1/2}$, where $\Sigma_p$ is a positive semi-definite matrix and $X$ is a $p \times n$ matrix with random *i.i.d.* entries. One of the main results from random matrix theory concerns the so-called spiked models, where $\Sigma_p$ is a low-rank perturbation of the identity matrix, namely $\Sigma_p = I_p + \sum_{i=1}^{k} \omega_i u_i u_i^\intercal$ with $k$ fixed with respect to $p, n$. (Baik et al., 2005) and (Paul, 2007) notably exhibited a phase transition phenomenon: as $p/n \to c$, if $\omega_i < \sqrt{c}$ the estimated principal component $\hat{u}_i$ using standard PCA is (almost surely) asymptotically orthogonal to the true principal component $u_i$ (*i.e.*, $\hat{u}_i^\intercal u_i \to 0$); on the other hand, if $\omega_i > \sqrt{c}$, $\liminf_n |\hat{u}_i^\intercal u_i| > 0$. This phase transition phenomenon has attracted recently much attention within the random matrix community (Benaych-Georges & Nadakuditi, 2011; Capitaine et al., 2009; Féral & Péché, 2007; Knowles & Yin, 2013).

The inconsistency of standard PCA in high dimensions motivated the idea to look for more structural information on the principal components. In particular, considering that the principal components

---

*Couillet's work is supported by the GSTATS UGA IDEX Datascience chair and the ANR RMT4GRAPH (ANR-14-CE28-0006).

are sparse in an appropriate basis (*e.g.*, in the wavelet domain), a large body of works have emerged and proposed improved PCA approaches that account for sparsity. One of the most consistent sparse PCA methods in the literature is the covariance thresholding (CT) algorithm (Krauthgamer et al., 2015). Based on the intuition that the small entries of the empirical covariance matrix $\hat{\mathbf{C}}$ induce noise in its principal components, this method consists in applying the popular *soft-thresholding* function (with threshold $\tau > 0$); $\mathrm{soft}(\,\cdot\,;\tau) : t \mapsto \mathrm{sign}(t) \cdot (|t| - \tau)_+$, entry-wise to the empirical covariance matrix $\hat{\mathbf{C}}$ and performing PCA on the resulting matrix. (Deshpande & Montanari, 2014; 2016) have theoretically demonstrated that the covariance thresholding algorithm recovers the sought-for principal components with high probability under controlled growth rates between $p$, $n$ and the sparsity level. In this paper, we show that the soft-thresholding method in fact falls within a broader class of kernel-based[1] PCA algorithms that are particularly suited to sparse PCA recovery. This method consists in considering the matrix $f(\hat{\mathbf{C}})$ instead of $\hat{\mathbf{C}}$ where $f$ is a function applied entry-wise. By imposing some constraints on $f$, most importantly that $f'(0) = f''(0) = 0$, sparse PCA can be performed with provably high accuracy for sufficiently large $n, p$.

The rest of the paper is organized as follows. In Section 2, we present the related work on sparse PCA. We recall some necessary concentration of measure tools and notions about sparse matrices in Section 3. Our main theoretical results are then provided in Section 4. Section 5 discusses the practical aspects and provides experimental results. Section 6 concludes the article.

*Notation:* In following, the notation $[n]$ denotes the set $\{1, \ldots, n\}$, $\lfloor a \rfloor$ denotes the integer part of $a$. Given a vector $x \in \mathbb{R}^n$, the $\ell_2$-norm of $x$ is denoted as $\|x\|^2 = \sum_{i=1}^n x_i^2$. Given an $n \times n$ matrix $M$, $M_{ij}$ or $[M]_{ij}$ denote the entry of the matrix $M$ at line $i$ and column $j$. $[M]_{.,j}$ denotes the $j$-th column vector of $M$ and $[M]_{i,.}$ its $i$-th line vector. The Frobenius norm $\|M\|_F$ of the matrix $M$ is defined as $\|M\|_F^2 = \sum_{i,j=1}^n M_{ij}^2$, and the operator norm $\|M\|_{op}$ of $M$ is defined as $\|M\|_{op} = \max_{\|x\|=1} \|Mx\|$. Finally, $\odot$ denotes the Hadamard product, with $[M \odot N]_{ij} = M_{ij} N_{ij}$.

## 2 RELATED WORK

The problem of sparse PCA has been tackled with a large range of techniques. Mainly, three classes of approaches emerge in the literature. Most popular techniques are optimization-based algorithms (d'Aspremont et al., 2005; Moghaddam et al., 2006; Zass & Shashua, 2007; Zou et al., 2006; Wright et al., 2009), where the idea is to see the problem of sparse PCA through an optimization perspective, and to propose methods to solve the latter by either considering a different formulation – *e.g.* semi-definite programming (SDP) or convex relaxations – or adding penalties to the original optimization problem such as a LASSO regularization. The second class of approaches covers matrix decomposition-based techniques (Asteris et al., 2014; Papailiopoulos et al., 2013; Shen & Huang, 2008), where sparse principal components are extracted through solving a low rank matrix approximation problem based on Singular Value Decomposition. Finally, most consistent sparse PCA methods adopt thresholding-based approaches: initial heuristics used factor rotation techniques and thresholding of eigenvectors to obtain sparsity (Cadima & Jolliffe, 1995). Based on the well-known power method, (Yuan & Zhang, 2013) introduced an efficient sparse PCA approximation to obtain the exact level of required sparsity, by truncating to zero the principal components iteratively except for their largest entries. A step further, under a spiked covariance model (see Section 4.1), (Ma et al., 2013) proposed a very efficient iterative thresholding approach for estimating principal subspaces in the sparse setting. Similarly, assuming a single-spike model, (Krauthgamer et al., 2015) proved that, when the sparsity level $s \geq \Omega(\sqrt{n})$, a standard SDP approach cannot recover consistently the sparse spike; in particular, the authors presented empirical results suggesting that for $s = \mathcal{O}(\sqrt{n})$, recovery is possible by a simple covariance thresholding algorithm. More recently, (Deshpande & Montanari, 2016) analyzed and theoretically proved, under a spiked model, that indeed the covariance thresholding algorithm (Krauthgamer et al., 2015) succeeds with high probability under controlled growth rates between $p, n$ and $s$.

In this work, while restricting ourselves to a setting where $p$ and $n$ grow at a controlled joint rate, we provide an elementary argument, based on a matrix-wise Taylor expansion controlled through a concentration of measure approach, that generalizes the CT method to a large family of kernel-based

---

[1] We use the *kernel-based* terminology to highlight that our work falls within the framework of kernel random matrices and should not be confused with the standard kernel PCA.

methods, by means of a kernel random matrix approach (El Karoui, 2010b;a). Concretely, we study kernel random matrices of the form $f(YY^\mathsf{T}/n)$ where $Y = \Sigma_p^{1/2} X$ and $X$ is a random matrix with $\mathcal{N}(0,1)$ *i.i.d.* entries. (El Karoui, 2010b) studied kernel matrices of the form $f(Y^\mathsf{T}Y/n)$ (*i.e.*, the so-called inner-product kernel matrices), which is equivalent to the case $\Sigma_p = I_p$ when considering the form $f(YY^\mathsf{T}/n)$. In particular, we elaborate from El Karoui's study by Taylor expanding $f(YY^\mathsf{T}/n)$ in the vicinity of $\Sigma_p$ entry-wise and controlling the resulting matrices via concentration arguments.

## 3 PRELIMINARIES

Before introducing our model setting we recall some definitions and notions of the concentration of measure theory (Ledoux, 2005) that are at the heart of our main results. Furthermore, we recall a definition, introduced by (El Karoui, 2008), of sparse matrices in the large-dimensional context that will also be exploited in this paper.

### 3.1 CONCENTRATION OF MEASURE RESULTS

We start by a definition of the notion of concentration for a real random variable.

**Definition 1** (Concentration of a Random Variable). *Given a function $\delta : \mathbb{R}_+ \to \mathbb{R}_+$, a random variable $Z$ is said to be $\delta$-concentrated (around its mean) and we write $Z \in \delta$, if for all $t > 0$, $\mathbb{P}\{|Z - \mathbb{E}Z| \geq t\} \leq \delta(t)$. In particular, $Z$ is said to be normally (resp., exponentially) concentrated when $\delta(t) = Ce^{-ct^2}$ (resp., $\delta(t) = Ce^{-ct}$) and we write $Z \in C\mathcal{N}(c\,\cdot)$ (resp., $Z \in C\mathcal{E}(c\,\cdot)$), where $C, c > 0$ are some absolute constants.*

In particular, $\delta$-concentration remains stable by application of Lipschitz functions:

**Proposition 1** (Concentration of Lipschitz Functions). *Given a $\lambda$-Lipschitz function $f : \mathbb{R} \to \mathbb{R}$ and a concentrated random variable $Z \in \delta$, we have $f(Z) \in \delta(\cdot/\lambda)$.*

As a consequence, linear combinations of $\delta$-concentrated random variables remain concentrated. However, products of $\delta$-concentrated random variables are more technical to handle, but we still have the following proposition in the case of normally concentrated random variables and which will be essential in this article.

**Proposition 2** (Square of Normally Concentrated Random Variables). *Given $Z \in C\mathcal{N}(c\,\cdot)$, the random variable $Z^2$ is exp-normally concentrated, precisely*

$$Z^2 \in K_C \mathcal{E}\left(\frac{c}{2}\,\cdot\right) + K_C \mathcal{N}\left(\frac{c}{16\,\mathbb{E}[Z]^2}\,\cdot\right), \tag{1}$$

*where $K_C > 0$ is a constant depending only on $C$.*

The extension of the notion of concentration to random vectors $Z \in \mathbb{R}^p$ demands that $\mathbb{R}^p \to \mathbb{R}$ Lipschitz functions are concentrated random variables.

**Definition 2** (Concentration of a Random Vector). *Given a function $\delta : \mathbb{R}_+ \to \mathbb{R}_+$ and a normal space $(E, \|.\|)$, a random vector $Z \in E$ is said to be $\delta$-concentrated if for any 1-Lipschitz function $f : E \to \mathbb{R}$, the random variable $f(Z)$ is $\delta$-concentrated. We note again $Z \in \delta$.*

In particular, we have the concentration of Gaussian random vectors in the sense of Definition 2 in the following proposition (Tao, 2012, Theorem 2.1.12).

**Proposition 3** (Normal Concentration of Gaussian Random Vectors). *A Gaussian vector $Z \in \mathbb{R}^p$, with independent and identically distributed $\mathcal{N}(0,1)$ entries, is normally concentrated independently on the dimension $p$. Furthermore, $Z \in 2\mathcal{N}(\cdot/2)$.*

**Remark 1.** *According to Definition 2, given a Lipschitz application $F : \mathbb{R}^p \to \mathbb{R}^q$ for $q \in \mathbb{N}^*$, Proposition 3 provides the normal concentration of all the random vectors $F(Z)$. In particular, note that our results are extensible to this family of vectors and random vectors with independent entries.*

## 3.2 $\varepsilon$-SPARSE MATRICES

When considering a large-dimensional random matrix setting, the notion of sparsity for such matrices is particularly attached to the choice of the matrix norm.[2] (El Karoui, 2008) introduced a definition ($\varepsilon$-sparsity) for sparsity of matrices that is compatible with spectral analysis, and specifically adapted to the operator norm. The $\varepsilon$-sparsity definition requires some notions from graph theory that we present in the following: to each $p \times p$ symmetric matrix $M$, we define its corresponding adjacency matrix as $\mathcal{A}(M) = \{\mathbb{1}_{M_{ij} \neq 0}\}_{i,j=1}^{p}$, which corresponds to a graph $\mathcal{G}_p$ with $p$ vertices. A walk is said to be closed on this graph if it starts and finishes at the same vertex and the number of edges traversed by a walk defines the length of this walk. Denote $\mathcal{C}_p(k)$ the set of closed walks of length $k$ on $\mathcal{G}_p$.

**Definition 3** ($\varepsilon$-sparse matrices (El Karoui, 2008, Definition 1)). *A sequence of covariance matrices $\{\Sigma_p\}_{p=1}^{\infty}$ is said to be $\varepsilon$-sparse if the sequence of their associated graphs $\{\mathcal{G}_p\}_{p=1}^{\infty}$ satisfies, for all $k \in 2\mathbb{N}$, $|\mathcal{C}_p(k)| \leq C_k \, p^{\varepsilon(k-1)+1}$ where $\varepsilon \in [0,1]$, $C_k > 0$ independent of $p$ and $|\mathcal{S}|$ denotes the cardinality of the set $\mathcal{S}$.*

The $\varepsilon$-sparsity is both useful and convenient to out study for the following reasons: 1) it is adapted to the analysis of the operator norm of large sparse matrices (as we give concentration results on the operator norm); 2) it is also more general than other sparsity notions such as in (Bickel & Levina, 2008). In the latter, the authors developed a natural permutation-invariant notion of sparsity which is more specific than Definition 3 as pointed out in the introduction of their article. Furthermore, note that both sparsity notions (Definition 3 and the one in (Bickel & Levina, 2008)) provide equivalent bounds for $\varepsilon < \frac{1}{2}$ and when considering the large dimensional $p \sim n$ setting (see subsection 2.4 in (Bickel & Levina, 2008)); this is precisely the setting considered in Corollary 2 introduced subsequently (*cf.* $\mu > 0$).

**Remark 2.** *As Definition 3 is based on a graph defined by its corresponding adjacency matrix, we have the following property: given an $\varepsilon$-sparse matrix $M$ and a function $f$ such that $f(0) = 0$ and $f(x) \neq_{x \neq 0} 0$, the matrix $f(M)$, resulting from the application of $f$ entry-wise to $M$, remains $\varepsilon$-sparse; this is simply a consequence of $\mathcal{A}(M) = \mathcal{A}(f(M))$.*

# 4 MAIN RESULTS

In this section, we first present the setting of the article. Then, we provide an asymptotic equivalent to the matrix $f(\hat{\mathbf{C}})$. Finally, we treat as a special case the application of our result to the context of sparse PCA.

## 4.1 GENERAL SETTING AND MAIN RESULTS

Consider a data matrix $Y \in \mathbb{R}^{p \times n}$ defined as

$$Y \equiv \Sigma_p^{1/2} X = (I_p + P)^{1/2} X, \tag{2}$$

where $X \in \mathbb{R}^{p \times n}$ is a random matrix with *i.i.d.* $\mathcal{N}(0,1)$ entries, $P = \sum_{i=1}^{k} \omega_i u_i u_i^{\mathsf{T}}$ and $U = [u_1, \dots, u_k] \in \mathbb{R}^{p \times k}$ is isometric. Here, $k$ refers to the number of principal components (or eigenvectors) $u_1, \dots, u_k \in \mathbb{R}^p$ to be evaluated, with $\omega_1 > \dots > \omega_k > 0$ the corresponding eigenvalues respectively. We define the quantity[3] $\beta_p \equiv \max_i \|[\Sigma_p^{1/2}]_{.,i}\|$.

**Assumptions:** There exists $B > 0$ independent of $p, n$ such that $\max_{ij} |[\Sigma_p]_{ij}| < B$. Besides, there exists $\epsilon > 0$ such that $\beta_p \leq B' \, n^{\frac{1}{4} - \epsilon}$ for all $p, n$ and for some absolute constant $B' > 0$.

Under these assumptions, our main technical result is as follows:

---

[2]Considering the identity matrix (which is a sparse matrix), $\|I_p\|_{op} = 1$ while $\|I_p\|_F = \sqrt{p} \to \infty$.

[3]The role of $\beta_p$ is to ensure the concentration of the quadratic form in equation 4 introduced subsequently. When $\Sigma_p$ is a sparse matrix, $\beta_p$ plays the same role as the maximum spike strength in the bounds given in (Deshpande & Montanari, 2016) for the CT method.

**Theorem 1** (Asymptotic Equivalent). *For $f$ a three-times continuously differentiable function, define the matrices $F$ and $\tilde{F}$ respectively by[4]*

$$F \equiv \left\{ f\left( \left[ \frac{1}{n} Y Y^\mathsf{T} \right]_{ij} \right) \right\}_{i,j=1}^p \;,\; \tilde{F} \equiv f(\Sigma_p) + \sum_{k=1}^2 \frac{f^{(k)}(\Sigma_p)}{k!} \odot \left[ \Sigma_p^{1/2} \left( \frac{1}{n} X X^\mathsf{T} - I_p \right) \Sigma_p^{1/2} \right]^{\odot k}.$$

*Then for $\eta > 0$ and for an absolute constant $C > 0$, we have with probability at least $1 - \eta$*

$$\|F - \tilde{F}\|_{op} \leq C \frac{\beta_p^6 \, p}{n^{3/2} \sqrt{\eta}}. \tag{3}$$

For a general smooth function $f$, the kernel random matrix $f(\hat{\mathbf{C}})$ is particularly difficult to analyze through the usual tools of random matrix theory, such as the moment or Stieltjes transform-based methods (Tao, 2012). Rather than directly analyzing such a kernel random matrix, Theorem 1 gives an asymptotic equivalent to it, in operator norm, that has mainly two properties. First, the approximation matrix $\tilde{F}$ contains "simple" objects that have already been analyzed in random matrix theory – in particular, the term $(X X^\mathsf{T}/n - I_p)$ in the expression of $\tilde{F}$. Second, the approximation in operator norm implies (by Weyl's inequality (Eisenstat & Ipsen, 1998, Theorem 4.1)) that, when $\|F - \tilde{F}\|_{op} \to 0$, $F$ and $\tilde{F}$ have the same eigenvalues and same "isolated" eigenvectors asymptotically (see Corollary 2 subsequently).

*Sketch of Proof of Theorem 1.* The main idea of the proof relies on the following intuition: for large $n$, the entries of $X X^\mathsf{T}/n - I_p$ and its successive Hadamard products tend to zero at controllable rate. The concentration of measure framework then allows for the control of non-linear functions of the entries of $X X^\mathsf{T}/n - I_p$. Of utmost importance to this end is the following lemma.

**Lemma 1** (A Concentration Result). *For all $i, j \in [p]$, the bilinear form $g_{ij}(X) \equiv [\Sigma_p^{1/2}]_{i,\cdot} \left( \frac{1}{n} X X^\mathsf{T} - I_p \right) [\Sigma_p^{1/2}]_{\cdot,j}$ satisfies*

$$g_{ij}(X) \in K\mathcal{E}\left( \frac{c_1 \, n}{\beta_p^2} \cdot \right) + K\mathcal{N}\left( \frac{c_2 \, n}{\beta_p^4} \cdot \right), \tag{4}$$

*for some absolute constants $c_1, c_2, K > 0$.*

*Proof.* Denoting by $v_i$ the $i$-th column vector of the matrix $\Sigma_p^{1/2}$, we have by the polarization identity, for all $M$ Hermitian, $v_i^\mathsf{T} M v_j = \frac{1}{4}[(v_i + v_j)^\mathsf{T} M (v_i + v_j) - (v_i - v_j)^\mathsf{T} M (v_i - v_j)]$. It thus suffices to prove the result for the quadratic form $g(X) = v^\mathsf{T} \left( \frac{1}{n} X X^\mathsf{T} - I_p \right) v$ where $v \in \mathbb{R}^p$. Noticing that $v^\mathsf{T} X X^\mathsf{T} v = \|v^\mathsf{T} X\|^2$ and $\mathbb{E}\left[ \frac{1}{n} v^\mathsf{T} X X^\mathsf{T} v \right] = v^\mathsf{T} v$, we need to prove the concentration of the random variable $\|v^\mathsf{T} X\|^2$. In fact, since $v^\mathsf{T} X$ is a Gaussian vector, by Proposition 3, $\|v^\mathsf{T} X\| \in 2\mathcal{N}\left( \frac{\cdot}{2\|v\|^2} \right)$ by Remark 1 and by Definition 2 since $M \mapsto v^\mathsf{T} M$ and $u \mapsto \|u\|$ are respectively $\|v\|$-Lipschitz and 1-Lipschitz functions. We get the final result by Proposition 2. $\qquad\square$

A Taylor expansion of $F$ around $f(\Sigma_p)$ then leads to controlling the operator norm of $f^{(3)}(\xi^n) \odot [\Sigma_p^{1/2}(X X^\mathsf{T}/n - I_p)\Sigma_p^{1/2}]^{\odot 3}$ for $\xi^n$ a matrix with entries in the set $[[Y Y^\mathsf{T}/n]_{ij}, [\Sigma_p]_{ij}]$ (or $[[\Sigma_p]_{ij}, [Y Y^\mathsf{T}/n]_{ij}]$). This follows precisely from exploiting Lemma 1 twice, to control the fluctuations of the entries of both $\xi^n$ (by the conditions on $\max_{ij} |[\Sigma_p]_{ij}|$ and $\beta_p$) and $[\Sigma_p^{1/2}(X X^\mathsf{T}/n - I_p)\Sigma_p^{1/2}]^{\odot 3}$, with the bound provided in the theorem statement, thereby completing the proof. $\qquad\square$

A detailed proof of Theorem 1 is provided in Section A.2 of the Appendix. From now on, to simplify our arguments, we make the following assumptions:
**Assumptions:** As $n \to \infty$,

**A1** $p/n \to c \in (0, \infty)$,        **A2** $\limsup_n \max_i \omega_i < \infty$; specifically $\limsup_n \beta_p < \infty$.

Under this setting, we have the following important corollary to Theorem 1.

---

[4] $f$ and $f^{(k)}$ are applied entry-wise and $\odot k$ stands for the element-wise $k$-th power.

**Corollary 1.** *Define the matrices $F$ and $\tilde{F}$ as in Theorem 1 and Assumptions **A1** and **A2** hold. Then, for $\eta > 0$*

$$F = \tilde{F} + \mathcal{O}_\eta(n^{-\frac{1}{2}}), \tag{5}$$

*where the notation $X = \mathcal{O}_\eta^m(n^{-\alpha})$ stands for the fact that $\mathbb{P}\left\{\|X\|_{op} \geq C\,n^{-\alpha}\,\eta^{-\frac{1}{2m}}\right\} \leq \eta$ for some absolute constant $C > 0$ and non-negative integer $m$.*

As a consequence of Corollary 1, we have, by the $\sin(\Theta)$ theorem of (Davis & Kahan, 1970), the corollary below concerning the eigenvectors of the matrices $F$ and $\tilde{F}$.

**Corollary 2.** *Let $v_1, \ldots, v_k$ and $\tilde{v}_1, \ldots, \tilde{v}_k$ denote respectively the $k$ principal eigenvectors of $F$ and $\tilde{F}$. Denote by $\Delta_i = \omega_i - \omega_{i+1}$ for $i \in [k-1]$. Then for $\eta > 0$, we have*

$$\max_{i\in[k]} \min_{s\in\{+1,-1\}} \Delta_i^2 \|v_i - s\tilde{v}_i\|^2 = \mathcal{O}_\eta(n^{-1}). \tag{6}$$

## 4.2 Special Case: Sparse PCA

To get an insight on our coming results, consider the scenario where $U$ contains finitely many non-zero entries. In this case, the perturbation matrix $P$ in Eq. equation 11 contains finitely many non-zero entries (say $s$) on each line and a simple enumeration shows that $|\mathcal{C}_p(k)| \leq p\,s^{k-1}$, thus $P$ is 0-sparse in the sense of Definition 3. Similarly, $I_p$ is 0-sparse and by the additive stability[5] of the $\varepsilon$-sparsity notion, $\Sigma_p$ remains 0-sparse. More generally, if we assume that it exists $\varepsilon \in [0, \frac{1}{2})$ such that the population covariance matrix $\Sigma_p$ is $\varepsilon$-sparse, we have the following set of consequences. By Corollary 1, choosing $f$ in such a way that $f'(0) = f''(0) = 0$ ensures that the terms $f'(\Sigma_p) \odot \ldots$ and $f''(\Sigma_p) \odot \ldots$ vanish in the expression of $\tilde{F}$. Indeed, for $k \in \{1, 2\}$

(i) Only finitely entries of $f^{(k)}(\Sigma_p)$ do not vanish, precisely by Remark 2, since $\mathcal{A}(f^{(k)}(\Sigma_p)) = \mathcal{A}(\Sigma_p)$,[6] the matrix $f^{(k)}(\Sigma_p)$ is also (almost) $\varepsilon$-sparse.

(ii) The matrix $F^{(k)} = \left[\Sigma_p^{1/2}\left(\frac{1}{n}XX^\intercal - I_p\right)\Sigma_p^{1/2}\right]^{\odot k}$ has entries of order $\mathcal{O}(n^{-k/2})$. As a result,[7] we have for $\eta > 0$ and for all $m > 0$, $\max_{i,j} |F_{ij}^{(k)}| = \mathcal{O}_\eta^m\left(n^{-\frac{k}{2}+\frac{1}{m}}\right)$.

Since in addition the operator norm of $\Sigma_p^{1/2}(XX^\intercal/n - I_p)\Sigma_p^{1/2}$ is typically of order $\mathcal{O}(1)$ (see *e.g.*, (Bai & Silverstein, 1998)), it is then easily seen that, for each $k \geq 1$, the operator norm of the Hadamard product $f^{(k)}(\Sigma_p) \odot F^{(k)}$ vanishes (see Lemma 5 in Appendix A). In particular, note that the non-zero entries of $\Sigma_p$ are controlled through the maximum entry of $F^{(k)}$ which is vanishing asymptotically, as mentioned in item (ii) above. On the opposite $f(\Sigma_p)$ does not vanish since it has entries bounded away from zero (as long of course as $f \neq 0$). We precisely have the following result.

**Theorem 2.** *Let $\mu > 0$ and suppose $\Sigma_p$ is a $\frac{1}{2+\mu}$-sparse matrix. For $f$ a three-times continuously differentiable function and for $\eta > 0$, we have for all $\epsilon \in (0, \frac{\mu}{2(3+2\mu)})$*

$$F = f(\Sigma_p) + \mathcal{O}_\eta^{\lfloor 1/\epsilon \rfloor}\left(n^{\frac{-\mu}{2(2+\mu)}+\epsilon\left(2-\frac{1}{2+\mu}\right)}\right) \quad s.t. \ f'(0) = f''(0) = 0. \tag{7}$$

*Proof.* See Section A.3 in Appendix A. (See Corollary 1 for the notation $\mathcal{O}_\eta^m(.)$.) ☐

**Remark 3.** *Theorem 2 gives a general result concerning the estimation of $\varepsilon$-sparse covariance matrices (more precisely, element-wise functionals of sparse covariance matrices). In particular, the spiked model in Eq. equation 11 with $U$ sparse corresponds to the particular case when $\mu \to \infty$; in this case, for $\eta > 0$ and for all $\epsilon \in (0, 1/4)$, $F = f(\Sigma_p) + \mathcal{O}_\eta^{\lfloor 1/\epsilon \rfloor}(n^{-\frac{1}{2}+2\epsilon})$.*

---

[5] See Fact .1 in (El Karoui, 2008).

[6] Given $M \in \mathbb{R}^{p\times p}$, its corresponding adjacency matrix is defined as $\mathcal{A}(M) = \{\mathbb{1}_{M_{ij}\neq 0}\}_{i,j=1}^p$.

[7] See proof of Lemma 5 in Appendix A for a proof of this result.

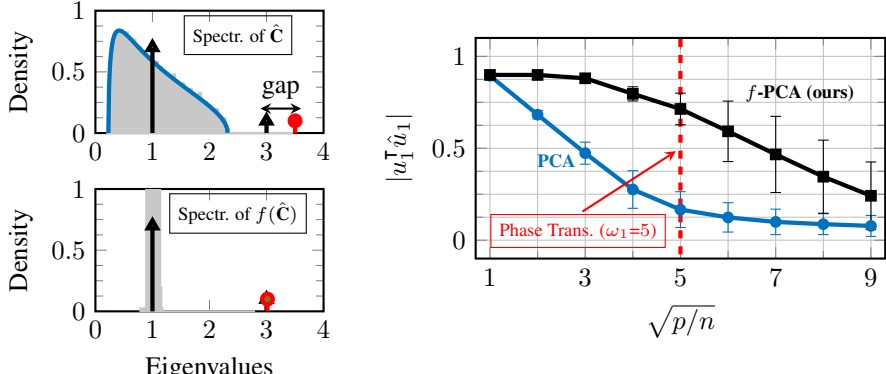

Figure 1: **(Left)** Spectrum of $\hat{\mathbf{C}}$ *(up)* and $f(\hat{\mathbf{C}})$ *(bottom)* for $p = 2048$ and $n = 7500$. Limiting Marčenko-Pastur density (Marčenko & Pastur, 1967) in blue versus spectrum of $\Sigma_p$ in black, with $\omega_1 = 2$; estimated largest eigenvalue in red. **(Right)** Alignment between estimated PC and GT (the "Three Peak" example of (Johnstone & Lu, 2009) in the "Symmlet 8" wavelet basis), in terms of $\sqrt{p/n}$. We considered $\omega_1 = 5$ and thus the phase transition for standard PCA occurs at $\sqrt{p/n} = 5$, thereby suggesting another phase transition for $f$-PCA. Curves obtained from 500 realizations of $X$.

One may then perform a PCA on $F$ for some function $f$ with $f'(0) = f''(0) = 0$ (we denote by $f^{1,2}(0) = 0$ these two conditions in the following). But, while $\Sigma_p = I_p + P$ is a low rank perturbation of the identity (therefore having only $k$ eigenvalues strictly greater than 1), $f(\Sigma_p)$ is likely more complex and not a mere low rank deformation of the identity. Now, if $\Sigma_p$ has all its non-zero entries greater than a certain threshold $\tau$, an appropriate choice for $f$ that avoids the deformation of $I_p + P$ is such that $f^{1,2}(0) = 0$ and $f(t) = t$ for all $|t| > \tau$.

Such a convenient choice is

$$f(t) = t(1 - e^{-at^2}), \qquad (8)$$

for some $a > 0$. This function notably satisfies

$$f'(t) = 1 + e^{-at^2}(2at^2 - 1) \Rightarrow f'(0) = 0,$$
$$f''(t) = -2ate^{-at^2}\left(2at^2 - 3\right) \Rightarrow f''(0) = 0.$$

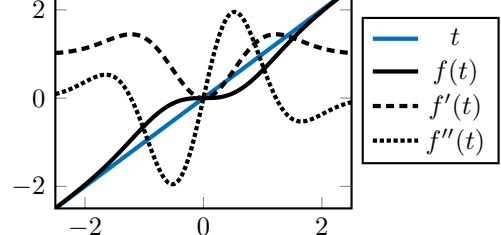

The figure above depicts the function $f$ along with its derivatives for $a = 1$. Note that a compromise in the choice of $a$ must be made that both maintains a close approximation of the identity by $f$ on a large range and rather small values of $f''$ in the vicinity of zero. Interestingly, it can be verified that the extrema of $f'$ are independent of $a$ but are found at $\pm\sqrt{\frac{3}{2a}}$ and thus smaller values of $a$ create sharper $f'$ in the vicinity of zero. Similarly, the extrema of $f''$ are found at $\pm\sqrt{\frac{3\pm\sqrt{6}}{2a}} \propto 1/\sqrt{a}$, and precisely given by the four values $2\sqrt{3a(3 \pm \sqrt{6})}e^{-\frac{1}{2}(3\pm\sqrt{6})} \propto \sqrt{a}$. Thus smaller $a$ induce larger maxima for $f''$ but no sharper slope.

## 5 EXPERIMENTS

In this section, we provide some experiments in the context of sparse PCA, where we consider the spiked model presented in Section 4.1. Precise setting given in caption of Figure 1. The spectrum of the sample covariance matrix (in gray) is quite different from that of $\Sigma_p$. One instead observes a "bulk" of eigenvalues spread in the vicinity of 1. Furthermore, one observes a gap between the true spike and the estimated spike (in red) through the sample covariance matrix. This phenomenon is well-understood in random matrix theory. In particular, the extreme eigenvalue in our setting converges almost surely to the quantity $(1 + \omega_1)\left(1 + \frac{c}{\omega_1}\right)$, where we recall that $c = \lim_n p/n$.

However, thanks to sparsity, the spectrum of $F = f(\hat{\mathbf{C}})$ closely matches that of $\Sigma_p$, as suggested by Theorem 2. In particular, the extreme eigenvalue, which corresponds to the principal component, is consistently estimated. Figure 1 (right) depicts the alignment between the estimated principal

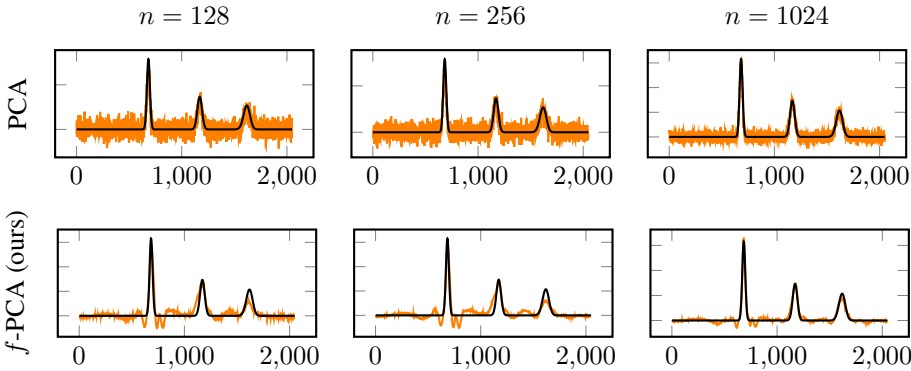

Figure 2: Principal component recovery (in orange) by standard PCA **(up)** and our method **(down)** for the "Three Peak" example of (Johnstone & Lu, 2009). The signal is sparse in the "Symmlet 8" wavelet basis. We use $p = 2048$, $\omega_1 = 5$ for the strength of the spike and different values of $n$.

component and ground truth, by standard PCA (in blue) and our method (in black), in terms of $\sqrt{p/n}$. Our method retrieves the principal component even when the spike is not visible in the spectrum of $\hat{\mathbf{C}}$; namely beyond the phase transition $\sqrt{p/n} \geq \omega_1$. In fact, the standard PCA result is too noisy compared with the one when considering $f(\hat{\mathbf{C}})$, as depicted in Figure 2. Further detailed examples are provided in Section A.4 of Appendix A, that confirm the consistency of the proposed method.

In terms of complexity, as our method consists in computing the sparse eigenvectors of a $p \times p$ matrix which can be done by power method, the complexity of estimating the principal component is about $\mathcal{O}(ps)$ where $s$ is the sparsity level. And regarding the performance *w.r.t.* state-of-the-art methods, Figure 3 depicts the performances of standard PCA, different state-of-the-art sparse PCA methods and our method, in terms of total projections score (left) and total projections error (right), for different values of the amplitudes $\omega_i$'s. We refer, in this figure, to standard PCA as PCA, TpowPCA for the method in (Yuan & Zhang, 2013), ITSPCA for the method in (Ma et al., 2013), CT refers to the method in (Deshpande & Montanari, 2016) and finally we refer to our method as $f$-PCA. The total projections score $\mathcal{S}$ and error $\mathcal{E}$ are given respectively by $\mathcal{S} = \frac{1}{k} \sum_{i=1}^{k} (u_i^\intercal \hat{u}_i)^2$ and $\mathcal{E} = \|UU^\intercal - \hat{U}\hat{U}^\intercal\|_F$, where $U = [u_1, \ldots, u_k]$ are the ground truth principal components and $\hat{U} = [\hat{u}_1, \ldots, \hat{u}_k]$ are the estimated ones.

As suggested theoretically and verified experimentally, our proposed method strongly attenuates the "noise component" of the sample covariance matrix and thus consistently estimates the principal components. In particular, in term of total projections score, PCA is the most inconsistent. In general, ITSPCA, CT and our method give equivalent results. The same holds when considering the total projections error as a metric, except that TpowPCA performs inconsistently, compared to PCA, for small values of amplitudes due to the initialization step from the PCA eigenvectors.

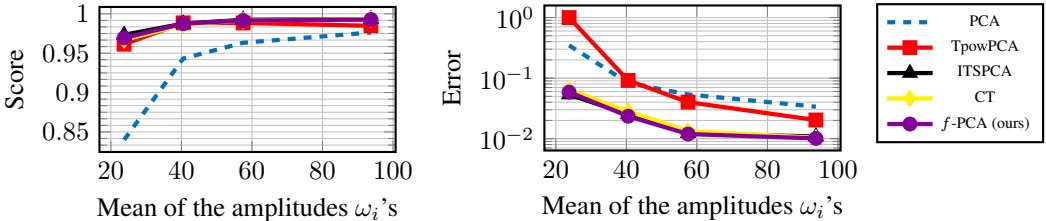

Figure 3: Performances of standard PCA, different state-of-the-art sparse PCA methods and our method in term of total projections score **(left)** and total projections error **(right)** for different values of the amplitudes $\omega_i$. The PCs $u_i$, for $i \in [4]$ are the "Three Peak", "Piece Poly", "Step New" and "Sing" signals of (Johnstone & Lu, 2009). We use $p = 2048$ and $n = 1024$. The soft-parameters $a$ and $\tau$ (respectively for our method and CT) are selected by cross-validation using a validation set of size $n$. The selected parameters are $a = 20$ and $\tau = 0.1$.

The mostly used concurrent methods to PCA in a sparse context are iterative truncated power methods (such as the TPower (Yuan & Zhang, 2013) algorithm or the ITSPCA (Ma et al., 2013) approach). These algorithms, despite great observed performances, as compared to standard PCA, suffer from two limitations. First, they are usually initialized from the PCA eigenvectors themselves and may not converge to good estimates. For weak signals, PCA is so impacted by noise that the mentioned initialization limitation may lead to non convergent or dramatically erroneous outcomes of the method. The proposed approach deals precisely with this limitation by strongly attenuating the "noise component" of the sample covariance matrix. In particular, our approach gives equivalent results to the CT method while generalizing it to the class of smooth functions $f$ such that $f'(0) = f''(0) = 0$, in the considered regime. The second limitation concerns the choice of the hyper-parameters; in fact, TPower and ITSPCA need to set up an arbitrary deterministic threshold value that maintains at each iteration step only most powerful components. The proposed method as well as CT need also to set up a "soft" parameter ($a$ and $\tau$ respectively). But, on the basis of (Cheng & Singer, 2013; Kammoun & Couillet, 2017), we believe that our present investigation can be extended to the *asymptotically non-trivial* setting where $\omega_i = \mathcal{O}(1/\sqrt{p})$ (in which case the dominant eigenmodes scale at a similar rate with residual noise); this setting may likely allow to exhibit and estimate optimal hyper-parameter choices. Notably, this setting has already been used in (Tiomoko Ali et al., 2018) in a different context, for hyper-parameters estimation.

## 6  CONCLUSION

In this paper, we tackled the problem of sparse PCA through a random matrix perspective thereby generalizing recent ideas to a broader kernel-based method. Our analysis of this problem has yielded insights into how the principal components can be consistently estimated. Namely, given a spiked covariance model $\hat{\mathbf{C}}$ and a smooth function $f$, we gave in this paper sufficient conditions on $f$ to consistently estimate the principal components through the matrix $f(\hat{\mathbf{C}})$. Our methodology can be generalized to other sparse covariance matrix-based contexts, in the same vein as the works in (Bickel & Levina, 2008; El Karoui, 2008).

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

# A    PROOFS AND FURTHER EXPERIMENTS

In this appendix we provide the proofs of the different results presented in the paper and some additional experiments that validate our findings. It includes the following items: (i) the proof of Theorem 1 (Section A.2); (ii) The proof of Theorem 2 concerning the analysis of the sparse case (Section A.3); and finally (iii) further experiments which confirm the consistency of our method and the necessity of the conditions $f'(0) = f''(0) = 0$ on the kernel function $f$, using the signals of Johnstone et al. (Johnstone & Lu, 2009) (Section A.4).

For convenience, we make the present appendix self-contained by recalling the preliminaries and the results presented in the main paper.

## A.1    PRELIMINARIES

**Proposition 2** (Square of Normally Concentrated Random Variables)**.** *Given $Z \in C\mathcal{N}(c \cdot)$, the random variable $Z^2$ is exp-normally concentrated, precisely*

$$Z^2 \in K_C \mathcal{E}\left(\frac{c}{2} \cdot\right) + K_C \mathcal{N}\left(\frac{c}{16 \, \mathbb{E}[Z]^2} \cdot\right), \tag{9}$$

*where $K_C > 0$ is a constant depending only on $C$.*

**Definition 2** (Concentration of a Random Vector)**.** *Given a function $\delta : \mathbb{R}_+ \to \mathbb{R}_+$ and a normal space $(E, \|.\|)$, a random vector $Z \in E$ is said to be $\delta$-concentrated if for any 1-Lipschitz function $f : E \to \mathbb{R}$, the random variable $f(Z)$ is $\delta$-concentrated. We note again $Z \in \delta$.*

**Proposition 3** (Normal Concentration of Gaussian Random Vectors (Tao, 2012, Theorem 2.1.12))**.** *A Gaussian vector $Z \in \mathbb{R}^p$, with independent and identically distributed $\mathcal{N}(0, 1)$ entries, is normally concentrated independently on the dimension $p$. Furthermore, $Z \in 2\mathcal{N}(\cdot/2)$.*

**Remark 1.** *According to Definition 2, given a Lipschitz application $F : \mathbb{R}^p \to \mathbb{R}^q$ for $q \in \mathbb{N}^*$, Theorem 3 provides the normal concentration of all the random vectors $F(Z)$.*

**Definition 3** ($\varepsilon$-sparse matrices (El Karoui, 2008, Definition 1))**.** *A sequence of covariance matrices $\{\Sigma_p\}_{p=1}^{\infty}$ is said to be $\varepsilon$-sparse if the sequence of their associated graphs[8] $\{\mathcal{G}_p\}_{p=1}^{\infty}$ satisfies, for all $k \in 2\mathbb{N}$*

$$|\mathcal{C}_p(k)| \leq C_k \, p^{\varepsilon(k-1)+1}, \tag{10}$$

*where $\varepsilon \in [0, 1]$, $C_k > 0$ independent of $p$ and $|\mathcal{S}|$ denotes the cardinality of the set $\mathcal{S}$.*

**Remark 2.** *As Definition 3 is based on a graph defined by its corresponding adjacency matrix, we have the following property: given an $\varepsilon$-sparse matrix $M$ and a function $f$ such that $f(0) = 0$ and $f(x) \neq_{x \neq 0} 0$, the matrix $f(M)$, resulting from the application of $f$ entry-wise to $M$, remains $\varepsilon$-sparse; this is simply a consequence of $\mathcal{A}(M) = \mathcal{A}(f(M))$.[1]*

---

[8]Defined through the corresponding adjacency matrix to $\Sigma_p$; given an $p \times p$ real symmetric matrix $M$, its corresponding adjacency matrix is defined as $\mathcal{A}(M) = \{\mathbb{1}_{M_{ij} \neq 0}\}_{i,j=1}^p$.

A.2 PROOF OF THEOREM 1

**Setting:** Consider a data matrix $Y \in \mathbb{R}^{p \times n}$ defined as

$$Y \equiv \Sigma_p^{1/2} X = (I_p + P)^{1/2} X, \tag{11}$$

where $X \in \mathbb{R}^{p \times n}$ with *i.i.d.* $\mathcal{N}(0,1)$ entries, $P = \sum_{i=1}^{k} \omega_i u_i u_i^\intercal$, $U = [u_1, \dots, u_k] \in \mathbb{R}^{p \times k}$ is isometric, $k$ refers to the number of principal components and $\omega_1 > \dots > \omega_k > 0$. Let $\beta_p \equiv \max_i \|[\Sigma_p^{1/2}]_{\cdot, i}\|$.

**Assumptions:** There exists $B > 0$ independent of $p, n$ such that $\max_{ij} |[\Sigma_p]_{ij}| < B$. Besides, there exists $\epsilon > 0$ such that $\beta_p \le B' \, n^{\frac{1}{4} - \epsilon}$ for all $p, n$ and for some absolute constant $B' > 0$.

Under these assumptions, we have

**Theorem 1.** *For $f$ a three-times continuously differentiable function, define the matrices $F$ and $\tilde{F}$ respectively by*

$$F \equiv f\left(\frac{1}{n} Y Y^\intercal\right) = \left\{ f\left(\left[\frac{1}{n} Y Y^\intercal\right]_{ij}\right) \right\}_{i,j=1}^{p}$$

$$\tilde{F} \equiv f(\Sigma_p) + \sum_{k=1}^{2} \frac{f^{(k)}(\Sigma_p)}{k!} \odot \left[\Sigma_p^{1/2}\left(\frac{1}{n} X X^\intercal - I_p\right)\Sigma_p^{1/2}\right]^{\odot k}.$$

*Then for $\eta > 0$ and for an absolute constant $C > 0$, we have with probability at least $1 - \eta$*

$$\|F - \tilde{F}\|_{op} \le C \, \frac{\beta_p^6 \, p}{n^{3/2} \sqrt{\eta}}. \tag{12}$$

*Proof.* Before starting the proof, we need to introduce the following key lemmas:

**Lemma 1** (A Concentration Result). *For all $i, j \in [p]$, the bilinear form $g_{ij}(X) \equiv [\Sigma_p^{1/2}]_{i, \cdot} \left(\frac{1}{n} X X^\intercal - I_p\right) [\Sigma_p^{1/2}]_{\cdot, j}$ satisfies*

$$g_{ij}(X) \in K\mathcal{E}\left(\frac{c_1 \, n}{\beta_p^2} \cdot\right) + K\mathcal{N}\left(\frac{c_2 \, n}{\beta_p^4} \cdot\right), \tag{13}$$

*for some absolute constants $c_1, c_2, K > 0$.*

*Proof.* Denoting by $v_i$ the $i$-th column vector of the matrix $\Sigma_p^{1/2}$, we have by the polarization identity, for all $M$ Hermitian, $v_i^\intercal M v_j = \frac{1}{4}[(v_i + v_j)^\intercal M(v_i + v_j) - (v_i - v_j)^\intercal M(v_i - v_j)]$. It thus suffices to prove the result for the quadratic form $g(X) = v^\intercal \left(\frac{1}{n} X X^\intercal - I_p\right) v$ where $v \in \mathbb{R}^p$. Noticing that $v^\intercal X X^\intercal v = \|v^\intercal X\|^2$ and $\mathbb{E}\left[\frac{1}{n} v^\intercal X X^\intercal v\right] = v^\intercal v$, we need to prove the concentration of the random variable $\|v^\intercal X\|^2$. In fact, since $v^\intercal X$ is a Gaussian vector, by Proposition 3, $\|v^\intercal X\| \in 2\mathcal{N}\left(\frac{\cdot}{2\|v\|^2}\right)$ by Remark 1 and by Definition 2 since $u \mapsto \|u\|$ and $M \mapsto v^\intercal M$ are respectively 1-Lipschitz and $\|v\|$-Lipschitz functions. We get the final result by Proposition 2. $\qquad\square$

**Lemma 2** (A Moment Result). *For $g_{ij}(X) \equiv [\Sigma_p^{1/2}]_{i, \cdot} \left(\frac{1}{n} X X^\intercal - I_p\right) [\Sigma_p^{1/2}]_{\cdot, j}$, we have, for all $k \in \mathbb{N}$ and for some absolute constant $C_k > 0$,*

$$\mathbb{E}|g_{ij}(X)|^{2k} \le C_k \, \frac{\beta_p^{4k}}{n^k}. \tag{14}$$

*Proof.* Given a random variable $Z$, we have

$$\forall m > 0, \; \mathbb{E}|Z|^m = \int_0^\infty m \, t^{m-1} \mathbb{P}\{|Z| \ge t\} dt,$$

whenever the right hand side is finite. Applying this identity to the random variable $g_{ij}(X)$ with $m = 2k$ and exploiting the concentration property in Lemma 1 yields the result. $\qquad\square$

The proof starts by a Taylor expansion of $F_{ij}$ in the vicinity of $[\Sigma_p]_{ij}$, *i.e.*,

$$F_{ij} = \sum_{k=0}^{2} \frac{f^{(k)}(\sigma_{ij})}{k!} F_{ij}^{(k)} + \frac{f^{(3)}(\xi_{ij}^n)}{6} F_{ij}^{(3)}$$

where $\sigma_{ij} = [\Sigma_p]_{ij}$, $\xi_{ij}^n \in \lfloor [YY^\intercal/n]_{ij}, \sigma_{ij} \rfloor$,[9] and $F^{(k)}$ is the matrix with entries

$$F_{ij}^{(k)} \equiv [\Sigma_p^{1/2}(n^{-1}XX^\intercal - I_p)\Sigma_p^{1/2}]_{ij}^k = g_{ij}(X)^k.$$

We have by Lemma 1 that $[YY^\intercal/n]_{ij}$ concentrates around $\sigma_{ij}$, so that $\xi_{ij}^n$ is bounded by $\sigma_{ij} + \varepsilon$, for all $\varepsilon > 0$, with high probability[10] (note that the condition $\max_{ij} |\sigma_{ij}| < B$ ensures that $\sigma_{ij}$ is bounded and the condition on $\beta_p$ ensures the quasi-exponential concentration of $[YY^\intercal/n]_{ij}$ around $\sigma_{ij}$; see considered Assumptions above), formally

$$\mathbb{P}\left\{|\xi_{ij}^n| \geq \sigma_{ij} + \varepsilon\right\} \leq \mathbb{P}\left\{|g_{ij}(X)| \geq \varepsilon\right\} \leq Ke^{-\frac{n}{\beta_p^2}\min(c_1\varepsilon, \frac{c_2\varepsilon^2}{\beta_p^2})}$$

$$\leq Ke^{-K'n^{\frac{1}{2}+2\epsilon}\min(c_1\varepsilon, K'c_2\varepsilon^2 n^{-\frac{1}{2}+2\epsilon})} \equiv p_n \to 0,$$

where $K' > 0$. And since $f^{(3)}$ is continuous, we deduce that $f^{(3)}(\xi_{ij}^n)$ is in particular bounded by

$$A \equiv \max_{x \in [\sigma_{ij}-\varepsilon, \sigma_{ij}+\varepsilon]} |f^{(3)}(x)|,$$

with probability $1 - p_n$. Knowing that the operator norm is bounded by the Frobenius norm, we look for a control of the Frobenius norm of the tailing term. We have

$$\|f^{(3)}(\xi^n) \odot F^{(3)}\|_F^2 \leq A^2 \|F^{(3)}\|_F^2. \tag{15}$$

By Lemma 2, for all $k \in \mathbb{N}$

$$\mathbb{E}\|F^{(k)}\|_F^2 = \sum_{i,j=1}^{p} \mathbb{E}\left[|g_{ij}(X)|^{2k}\right] \leq C_k \frac{p^2\beta_p^{4k}}{n^k},$$

for some absolute constant $C_k > 0$. Thus, by *Markov's inequality*, we have for all $\eta > 0$

$$\mathbb{P}\left\{\|F^{(k)}\|_F \geq \frac{p\beta_p^{2k}}{n^{\frac{k}{2}}}\sqrt{\frac{C_k}{\eta}}\right\} \leq \eta.$$

Recalling Eq. equation 15, we have with probability at least $1 - \eta$

$$\|f^{(3)}(\xi^n) \odot F^{(3)}\|_F \leq C\frac{p\beta_p^6}{n^{\frac{3}{2}}\sqrt{\eta}}.$$

## A.3 Proof of Theorem 2

$\square$

**Assumptions:** As $n \to \infty$,

    **A1** $p/n \to c \in (0, \infty)$.

    **A2** $\limsup_n \max_i \omega_i < \infty$; specifically $\limsup_n \beta_p < \infty$.

With these assumptions, we have the following corollary to Theorem 1.

**Corollary 1.** *Define the matrices $F$ and $\tilde{F}$ as in Theorem 1 and let Assumptions A1 and A2 hold. Then, for $\eta > 0$*

$$F = \tilde{F} + \mathcal{O}_\eta(n^{-\frac{1}{2}}), \tag{16}$$

*where the notation $X = \mathcal{O}_\eta^m(n^{-\alpha})$ stands for the fact that $\mathbb{P}\left\{\|X\|_{op} \geq Cn^{-\alpha}\eta^{-\frac{1}{2m}}\right\} \leq \eta$ for some absolute constant $C > 0$ and non-negative integer $m$.*

---

[9]The notation $\lfloor a, b \rfloor$ stands for the interval $[a, b]$ if $a < b$ or $[b, a]$ otherwise.

[10]For a given asymptotic variable $n$, we say that an event $E_n$ occurs with high probability when it exist a function $\psi(n)$ quasi-exponentially decreasing in $n$ such that $\mathbb{P}\{E_n\} \geq 1 - \psi(n)$.

**Theorem 2.** *Let $\mu > 0$ and suppose $\Sigma_p$ is a $\frac{1}{2+\mu}$-sparse matrix. For $f$ a three-times continuously differentiable function and for $\eta > 0$, we have for all $\epsilon \in (0, \frac{\mu}{2(3+2\mu)})$*

$$F = f(\Sigma_p) + \mathcal{O}_\eta^{\lfloor 1/\epsilon \rfloor}\left(n^{\frac{-\mu}{2(2+\mu)}+\epsilon\left(2-\frac{1}{2+\mu}\right)}\right) \ \text{s.t.} \ f'(0) = f''(0) = 0. \tag{17}$$

*Proof.* The proof needs the introduction of the following two lemmas, that can be found in (El Karoui, 2008, Lemma A.1 and A.2) and which are a consequence of the $\varepsilon$-sparsity notion[11]

**Lemma 3.** *Given an $\varepsilon$-sparse $p \times p$ real symmetric matrix $M$ and calling $m = \max_{ij} |M_{ij}|$, we have, for all $k \in 2\mathbb{N}$*

$$\|M\|_{op} \leq \text{trace}(M^k)^{1/k} = \mathcal{O}(m\, p^{\varepsilon(1-1/k)+1/k}). \tag{18}$$

**Lemma 4.** *Given two real symmetric matrices $M$ and $N$ with $|M_{ij}| \leq N_{ij}$. Then, we have $\|M\|_{op} \leq \|N\|_{op}$.*

First, we show that when $\Sigma_p$ is $\varepsilon$-sparse, the Hadamard product $f^{(k)}(\Sigma_p) \odot F^{(k)}$ is of vanishing operator norm for $k \geq 1$, precisely

**Lemma 5.** *Let $\mu > 0$, suppose $\Sigma_p$ is a $\frac{1}{2+\mu}$-sparse matrix. For $f$ a real and differentiable function, $k \in \{1, 2\}$ such that $f^{(k)}(0) = 0$ and for $\eta > 0$, we have for all $\epsilon \in (0, \frac{k(2+\mu)-2}{2(3+2\mu)})$*

$$\|f^{(k)}(\Sigma_p) \odot F^{(k)}\|_{op} = \mathcal{O}_\eta^{\lfloor 1/\epsilon \rfloor}\left(n^{\frac{2-k(2+\mu)}{2(2+\mu)}+\epsilon\left(2-\frac{1}{2+\mu}\right)}\right).$$

*Proof.* We start by proving that the matrix $F^{(k)}$ has entries of order $\mathcal{O}(n^{-k/2})$. In fact, we have by Lemma 2, for all $m \in \mathbb{N}^*$

$$\mathbb{E}|F_{ij}^{(k)}|^{2m} = \mathbb{E}|g_{ij}(X)|^{2km} = \mathcal{O}(n^{-km}),$$

thus applying *Markov's inequality* to the random variable $|F_{ij}^{(k)}|^{2m}$ yields to the following tail control.

$$\mathbb{P}\{|F_{ij}^{(k)}| \geq t\} \leq \frac{\mathbb{E}|F_{ij}^{(k)}|^{2m}}{t^{2m}} \leq C\,n^{-km}\,t^{-2m},$$

where $C$ is an absolute constant. Recalling Assumption **A1** and by the union bound, we have

$$\mathbb{P}\{\max_{ij}|F_{ij}^{(k)}| \geq t\} \leq \sum_{i,j=1}^{p} \mathbb{P}\{|F_{ij}^{(k)}| \geq t\} \leq p^2\,\mathbb{P}\{|F_{ij}^{(k)}| \geq t\} \leq C\,n^{2-km}\,t^{-2m},$$

which implies for $\eta > 0$ and for all $m > 0$

$$\max_{ij}|F_{ij}^{(k)}| = \mathcal{O}_\eta^m\left(n^{-\frac{k}{2}+\frac{1}{m}}\right) \tag{19}$$

Besides, let $M$ be the matrix defined as $M \equiv \max_{ij}|F_{ij}^{(k)}| \cdot f^{(k)}(\Sigma_p)$, we have

$$|[f^{(k)}(\Sigma_p) \odot F^{(k)}]_{ij}| \leq M_{ij},$$

thus, one has by Lemma 4

$$\|f^{(k)}(\Sigma_p) \odot F^{(k)}\|_{op} \leq \|M\|_{op} = \max_{ij}|F_{ij}^{(k)}| \cdot \|f^{(k)}(\Sigma_p)\|_{op}.$$

In particular, since $f^{(k)}(\Sigma_p)$ is $\frac{1}{2+\mu}$-sparse (by Remark 2), we have by Lemma 3 and by equation 19, for some $\eta > 0$

$$\|f^{(k)}(\Sigma_p) \odot F^{(k)}\|_{op} = \mathcal{O}_\eta^{2m}\left(n^{\frac{1}{2+\mu}(1-\frac{1}{2m})+\frac{1}{2m}-\frac{k}{2}+\frac{1}{2m}}\right),$$

choosing $\epsilon = \frac{1}{2m} < \frac{k(2+\mu)-2}{2(3+2\mu)}$ yields the final result. $\qquad\square$

When considering $f$ such that $f'(0) = f''(0) = 0$, the result holds by Corollary 1 and Lemma 5. In fact, the dominant order corresponds to $k = 1$ in Lemma 5. Which completes the proof. $\qquad\square$

---

[11]Through the identity $\text{trace}(M^k) \leq \max_{ij}|M_{ij}|^k \cdot |\mathcal{C}_p(k)|$.

### A.4 FURTHER EXPERIMENTS

### A.4.1 HIGHER RANK CASE

In this section, we provide further experiments by considering a rank three case and by using the "Three Peak", "Piece Poly" and "Step New" signals of Johnstone et al. (Johnstone & Lu, 2009), in the "Symmlet 8" wavelet basis, as principal components. We compare the estimated PCs by our method with the kernel function in equation 8 to the estimated ones through standard PCA and the CT method (Deshpande & Montanari, 2016). As shown in Figure 4, the proposed method retrieves consistently the principal components compared to a standard PCA. In particular, we obtain results that are similar to the ones obtained by the CT method while generalizing it to the class of smooth functions with $f'(0) = f''(0) = 0$.

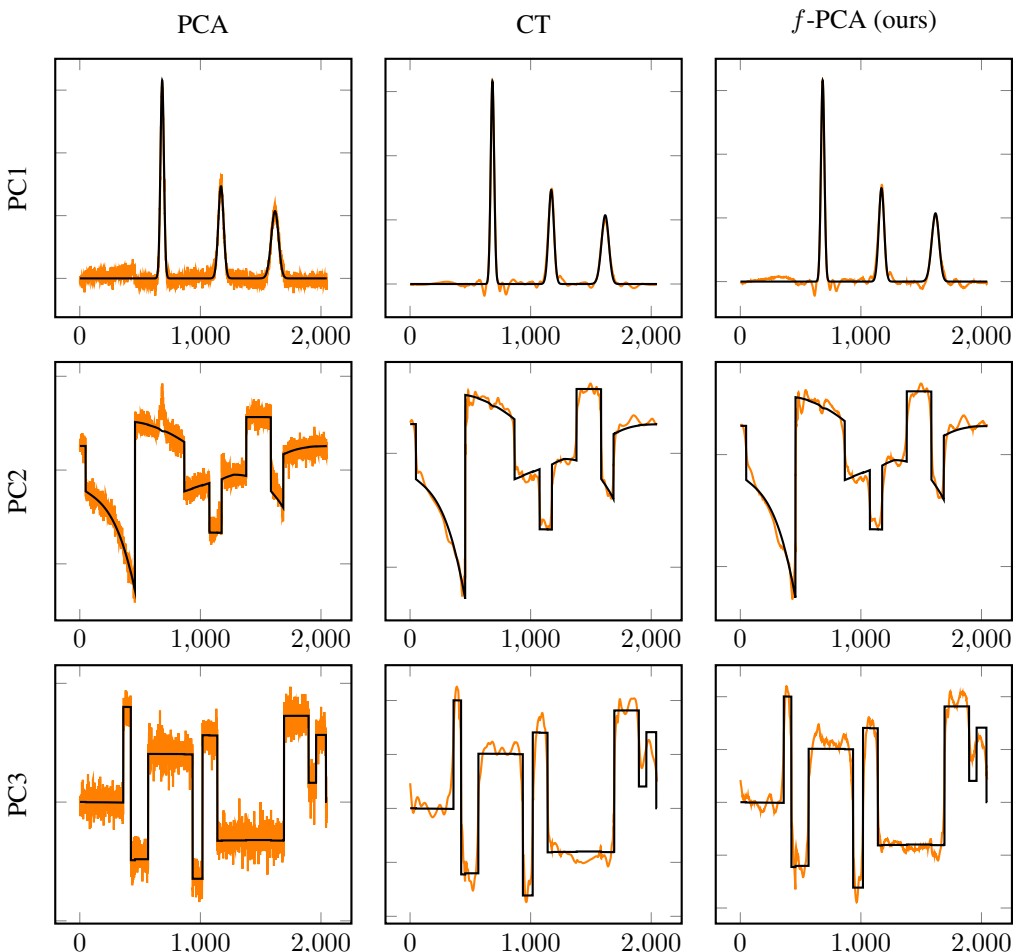

Figure 4: Multiple principal components ($k = 3$) recovery (in orange) with standard PCA **(left)**, the CT method **(middle)** and our method **(right)** where the PCs are considered to be the "Three Peak", "Piece Poly" and "Step New" signals of (Johnstone & Lu, 2009), in the "Symmlet 8" wavelet basis. We use $p = 2048$, $n = 1024$ and the spikes strengths are set respectively as $\omega_1 = 100$, $\omega_2 = 75$ and $\omega_3 = 50$. In particular, we note the similarity between the results obtained by our method and CT.

### A.4.2 OTHER CHOICES OF THE KERNEL FUNCTION $f$

In this section, we consider functions of the form $f(t) = \alpha t^3 + \beta t^2 + \gamma t$ where $\alpha, \beta, \gamma \in \mathbb{R}$ are some parameters to fix in order to allow or not the conditions $f'(0) = f''(0) = 0$. In particular, we set different parameters choices for $\alpha, \beta$ and $\gamma$ in order to validate these conditions. Figure 5 depicts different PC recovery using the $f$-PCA method with the considered class of functions. As we can observe from this figure, the "cleanest" signal recovery is obtained when $\alpha \neq 0, \beta = 0, \gamma = 0$ (*i.e.*, when $f'(0) = f''(0) = 0$) thereby validating our theoretical conditions on the kernel function $f$ for a consistent sparse PCA recovery. Note that these conditions are necessary but not sufficient in the sense that $f$ has to be linear for large values of $t$ (In particular, this is the case for the function $f$ given by equation 8). In fact, the outcome provided by $f$-PCA for $f(t) = \alpha t^3$ with $\alpha \neq 0$ is not optimal as the obtained signal is deformed (due to the unverified linearity condition), compared to the GT one.

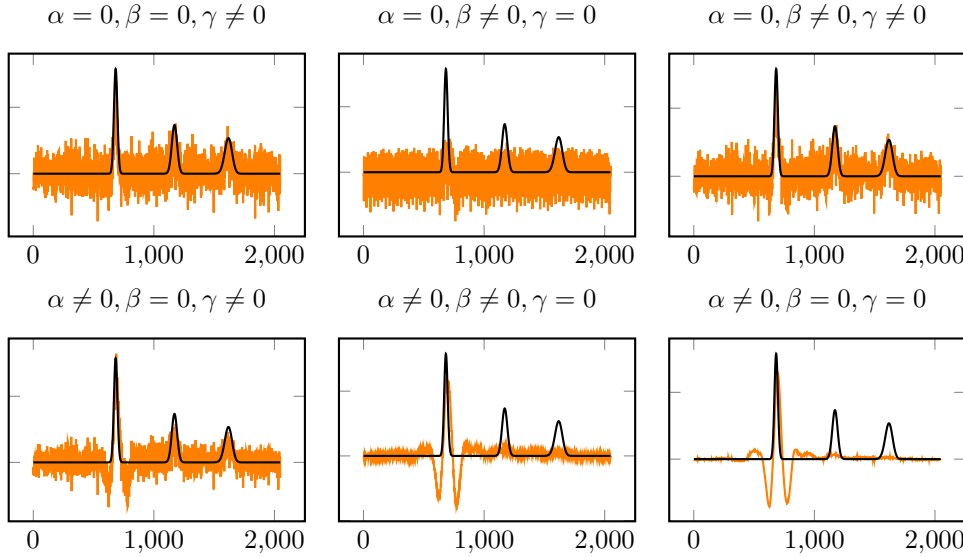

Figure 5: PC recovery (in orange) by $f$-PCA with the function $f(t) = \alpha t^3 + \beta t^2 + \gamma t$ for different values of the parameters $(\alpha, \beta, \gamma) \in \mathbb{R}^3$. We consider the "Three Peak" example of (Johnstone & Lu, 2009) which is sparse in the "Symmlet 8" wavelet basis. We use $p = 2048$, $n = 256$ and $\omega_1 = 5$. In particular, we notice that the "cleanest" signal is obtained when $\alpha \neq 0, \beta = 0, \gamma = 0$ which validate our theoretical conditions $f'(0) = f''(0) = 0$.

