# OpenReview forum: "A Kernel Random Matrix-Based Approach for Sparse PCA"
_ICLR.cc/2019/Conference_

### Official Review · AnonReviewer2 · 2018-11-02
**A kernel matrix-based algorithm for PCA**

**Rating:** 6
**Confidence:** 4

**Review:**

This paper proposes an algorithm to approximate kernel matrix based on the Taylor expansion of the element-wise functions. The authors provide a spectral norm based error bound for their method and the corresponding results for the special case, \epsilon-sparse matrix.

I have some comment as follow.

1. Can you provide some comparison with Nystrom methods? It is very popular for kernel approximation and looks more efficient than the proposed algorithm.

2. The analysis relies on the Gaussian assumption on the input matrix. Can we extend it to more general case?

3. In section 5, the paper said “as our method consists in computing the sparse eigenvectors of a p \times p matrix which can be done by power method, the complexity of estimating the principal component is about O(ps) where s is the sparsity level”. The time complexity of the proposed algorithm is not clearly.
a) Is there any bound for the sparsity level s? Why the eigenvectors of p \times p matrix is sparse?
b) The convergence of power method is heavily affected by the eigen-gap of the matrix. Is there any theoretical or empirical result for the convergence behavior of power method on approximate matrix and original matrix?

---

> ### Author Response · Authors · 2018-11-08
> **Response to Reviewer2**
>
> We thank the reviewer for the time spent reviewing our work. We believe that there might be a slight misunderstanding in the main consequences of our findings. Indeed, our work does not propose a low rank approximation of kernel matrices per se. The approximation matrix given in Theorem 1 is only useful from a mathematical standpoint, which, by leading up to a mathematically tractable spiked model (Eq. (3)), allows for a deep and thorough asymptotic analysis of the matrix F. In particular, Theorem 1 allows us to show that a careful choice of the function f (namely with f’(0)=f’’(0)=0) leads to an accurate method for estimating the principal components in a sparse context. As a consequence, we find here a necessary and sufficient condition for generic functions f applied entry-wise on the sample covariance matrix to retrieve sparse principal component, thereby largely generalizing the so-far uniquely known case of the soft-thresholding function (which has all its derivatives null at zero).
>
> In the following, we provide answers to the detailed review comments:
>
> 1. As we pointed out above, the proposed approach falls within the sparse PCA problem which is different from the kernel approximation problem and, thus, we are not aiming here at a numerically improvement of PCA but at an actual improvement of PCA for sparse eigenvectors.
> 2. To simplify our arguments, we have chosen to work under the Gaussian assumption. However, our results are extensible to more general cases, particularly to the case of concentrated vectors (Lipschitz-ally transformed Gaussian vectors or random vectors with independent entries) which is a consequence of Remark 1.
> 3.a) In this work, we have considered a sparsity level s = O(1) which corresponds to the case e = 0 from the e-sparsity notion perspective. As the main purpose of our paper is to provide the conditions under which the PCs can be consistently recovered from a smoothly transformed sample covariance matrix, providing bounds for the sparsity level is out of the scope of the proposed findings. However, note that we consider in Theorem 2 the estimation of element-wise functionals of e-sparse covariance matrices with e >= 0, which implicitly suggests that our results are still valid for s = O(n^e).
> 3.b) As we have mentioned in point 3.a), the aim of the paper is to determine the necessary conditions for f to powerful sparse PCA recovery and our results guarantee the approximation of f(actual covariance) through f(sample covariance). In particular, a power method can be used to fast compute the (dominant) eigenvectors of f(actual covariance).

---

### Official Review · AnonReviewer3 · 2018-11-02
**Good paper, with some issues in notations**

**Rating:** 7
**Confidence:** 2

**Review:**

This paper discusses the sparse PCA problem from the random matrix perspective. First, it establishes a theorem that connect a f(sample covariance) to f(actual covariance) in Theorem 1, and shows that when f if three-times continuous differentiable,  f(sample covariance) can be written in terms of actual covariance and $1/n XX^T-I$. Then, based on this theorem, it shows that if f'(0)=f''(0)=0, then f(sample covariance) is well approximated by f(actual covariance).

Based on this result, a procedure for sparse PCA is proposed: first, soft threshold the sample covariance (the thresholding function is described in (9)); second, calculate the top eigenvectors of the thresholded sampled covariance. In simulations, it has similar performance as some popular sparse PCA algorithms.

While I think the result of the paper is certainly interesting and worth publication, many notations in the papers are not clear and I can not verify the proofs completely as a result. For example:
1. Z\in C E(c,.) in Definition 1---what is the set E(c,.) and is it the same as N(c,.) as implied in definition 1? But it E(c,.) is the same as N(c,.), why they are treated differently in Proposition 2?


2.  the subscript {.,i} in the first paragraph Section 4.1 (I guess it means the i-th column).

3. how are f and f^{(k)} defined for matrices in Theorem 1 and what is the supscript {\odot k}--elementwisely k-th power?

4. The notation O_\eta in (8).

Some other thought: is the assumption before Theorem 1 reasonable? Can the author(s) add some comments and show that it holds for a reasonable Sigma=I+P: the assumption and Theorem 1 would hold for very small P, but that is not an interesting case.

The function in (9) is essentially a soft-threshoding procedure. Can the method in this work be used to prove other thresholding procedures such as hard thresholding?

---

> ### Author Response · Authors · 2018-11-08
> **Response to Reviewer3**
>
> We thank the reviewer for his time reviewing our work and for his pertinent and constructive comments on our paper.
>
> The reviewer discusses the unclarity of some notations that we used through the paper. We respond to each point concerning this aspect in the following:
>
> 1. The notation Z \in C N(c,.), defined in the article, means that the random variable Z satisfies the concentration property P(| Z - E(Z) | > t) < C e^{-ct^2} where C and c are absolute constants and E(Z) stands for the expectation of Z. The same holds for Z \in C E(c,.) etc. We merely used these notations to avoid repeating the formulas P(| Z - E(Z) | > t) < C e^{-ct^2}.
> 2. The subscript {.,i} indeed stands for the i-th column. We have added an explanation for the used notations in page 2 before section 2.
> 3. f and f^{(k)} are applied element-wise to the actual covariance matrix, in particular, the subscript {\odot k} stands for the element-wise k-th power. We will add clarifications about these notations in the updated article.
> 4. We have introduced the notation O_\eta in Corollary 1, but will be recalled in Theorem 2 for clarification.
>
> The reviewer also discusses the assumption before Theorem 1. Note that when P is sparse (which is the scenario of interest for the sparse PCA problem) and when the signal-to-noise-ratio is finite (cf. assumption A2) then beta_p = O(1). As such, the assumptions and Theorem 1 are reasonable as our main interest lies in the sparse PCA setting.
>
> Finally, the reviewer highlights the use of our proposed method to other thresholding procedures. Note that the proposed approach imposes that the key features of f, to attenuate the noise component of the sample covariance, lie in its local behavior at zero. So one can use other thresholding procedures (with different behaviors far away from zero), depending on the application, with only the soft constraints f’(0)=f’’(0)=0 on f.

---

### Official Review · AnonReviewer1 · 2018-11-06
**Motivation and contribution to be narrowed and clarified**

**Rating:** 5
**Confidence:** 5

**Review:**

The paper aims at generalizing a result by Deshpande and Montanari.

Deshpande and Montanari prove a result for the covariance thresholding algorithm which consists of (1) removing noise from an empirical covariance matrix of a (single) spiked sparse PCA model using soft thresholding, then (2) computing the leading eigenvector of the denoised matrix and finally (3) picking the leading coordinates of the leading eigenvector.

The current paper focuses on step (1) in the above mentioned process. They claim to generalize the use of soft thesholding to more general functions applied element-wise to the empirical covariance matrix.

There is a questionable term in the paper's title: the word "kernel" is mistakenly used. In fact a symmetric matrix is the Gram matrix of a kernel if it is positive semi-definite. The empirical covariance is PSD, but when you apply a function to it elementwise it has no guarantee of conserving the PSD property.

Regarding the motivation of the paper (1) the paper claims to study the rank K case with arbitrary K>=1. Deshpande and Montanari studied K=1 and the analysis is already an interesting result. Generalizing it to higher ranks poses many questions, including the sparse vector supports' overlaps.
It is not clear to me why authors insist on trying to generalize the result to a function f that has broader properties. The main motivation is to recover the support of the (leading) sparse eigenvector / PC. It should not be to try denoising the empirical covariance with a complicated function f. If the focus is on generalizing the soft-thresholding part of the approach, then the real question can be formulated as what is the optimal f given that we have this or that property in the data? This often leads to Bayesian analysis of the problem.

The numerical experiments do not show any substantial improvement obtained using the prescribed method over using the baseline method (covariance thresholding). I suspect that authors can emphasis the benefit of their method by picking f to hold certain properties that reflect the noise process and beat covariance thresholding in those regimes.
Figure 1 right hand side. I do not see why authors refer to phase transition. I don't see a phase transition happening there.

---

> ### Author Response · Authors · 2018-11-08
> **Response to Reviewer1**
>
> We first thank the reviewer for the time taken to review our paper and for his pertinent and constructive comments.
>
> As hinted by the reviewer, the aim of our study is to generalize the use of soft thresholding to a broader class of functions. More importantly, we show in this paper how sparse principal components (PCs) can be consistently estimated through element-wise application of a smooth function f to the sample covariance matrix (precisely, we find that the necessary conditions on f are f’(0)=f’’(0)=0, while soft thresholding has all its derivatives null at zero).
>
> We argue that our method gives equivalent results to the covariance thresholding method (in the sparse PCA context). However, our approach is constructive, in the sense that it establishes the necessary conditions on f to consistently estimate the PCs, unlike the soft thresholding method which was proposed as a heuristic initially by (Krauthgamer et al., 2015).
>
> The reviewer questioned about the term “kernel” in our paper’s title and we argue that it is questionable. Note that we have used this terminology (namely, the term “kernel random matrix”) to highlight that our work falls within the framework of kernel random matrices (Cheng & Singer, 2013; El Karoui, 2010). Indeed, as mentioned by the reviewer, the resulting matrix after applying a function element-wise to the sample covariance may not result in a positive semi-definite matrix.
>
> Concerning the reviewer’s comment about the case of rank K with K>=1, it should be clarified that this point is not the scope of the paper since we do not focus on the decidability problem that already occurs even if P were perfectly known.
>
> As discussed above, our approach provides insights about the behavior of element-wise functionals of the sample covariance matrix for a wide class of functionals. Our subsequent step, as mentioned in the conclusion, is precisely to determine optimal choices of f, depending on the underlying scenario. This will be accessible through a study “near” the phase-transition where sparse PCA becomes a hard task; there, a more thorough comparison of the residual noise (there having a semi-circle spectrum) and the signal information can be performed similar to previous works in related settings (Cheng & Singer, 2013; Couillet & Benaych, 2016). Yet this analysis demands quite elaborate tools, currently under investigation, which the present “introductory” article does not cover so far.
>
> On this aspect, the reviewer raises the fact that our proposed method leads to equivalent results to the covariance thresholding method in the sparse PCA context, under ideal parameter settings. Assuming, as we believe, that our subsequent study will provide means to properly select those ideal parameters, the resulting findings will allow for a more efficient and flexible ultimate sparse-PCA algorithm (where the choice of the soft-thresholding parameter is made harder by the uneasy theoretical analysis of the non-differentiable soft-thresholding function). Indeed, the continuously derivability property on the kernel function f is a necessary property to address the theoretical analysis of the spectral norm of f(sample covariance) as hinted in Theorem 1.4. in [Fan et al.], note that a quadratic interpolation of the soft-thresholding function is considered in [Fan et al.] to satisfy this property, thereby highlighting the importance of using a smooth kernel function f as we propose.
>
> Finally, concerning the remark about Figure 1 (right), note that we have illustrated the phase transition for standard PCA which occurs at $\omega = sqrt( p / n)$, which suggests that our method performs efficiently beyond this phase transition. We will clarify this aspect in the updated version of the article to avoid misunderstanding.
>
> [Fan et al.] Zhou Fan and Andrea Montanari. "The spectral norm of random inner-product kernel matrices." Probability Theory and Related Fields (2015): 1-59.

---

### Meta-Review · Area_Chair1 · 2018-12-13

**Confidence:** 3
**Recommendation:** Accept (Poster)

**Metareview:**

The manuscript studies a random matrix approach to recover sparse principal components. This work extends prior work using soft thresholding of the sample covariance matrix to enable sparse PCA. In this light, the main contribution of the paper is a study of generalizing soft thresholding to a broader class of functions and showing that this improves performance. The contributions of this paper are primarily theoretical.

The reviewers and AC note issues with the discussion that can be further improved to better illustrate contributions, and place this work in context. In particular, multiple reviewers assumed that "kernel" referred to the covariance matrix. The authors provide a satisfactory rebuttal addressing these issues.

While not unanimous, overall the reviewers and AC have a positive opinion of this paper and recommend acceptance.